# Cytotoxic Steroidal Saponins Containing a Rare Fructosyl from the Rhizomes of *Paris polyphylla* var. *latifolia*

**DOI:** 10.3390/ijms24087149

**Published:** 2023-04-12

**Authors:** Tian-Yi Li, Yang Du, Min-Chang Wang, Ke Liu, Yang Liu, Yu Cao, Yuan-Yuan Wang, Wen-Wen Chen, Xiao-Ying Qian, Peng-Cheng Qiu, Hai-Feng Tang, Yun-Yang Lu

**Affiliations:** 1Department of Chinese Materia Medica and Natural Medicines, School of Pharmacy, Air Force Medical University, Xi’an 710032, China; 2Xi’an Modern Chemistry Research Institute, Xi’an 710065, China

**Keywords:** *Paris polyphylla* var. *latifolia*, fructosyl, spirostanol saponins, apoptosis, network pharmacology

## Abstract

A phytochemical investigation of the steroidal saponins from the rhizomes of *Paris polyohylla* var. *latifolia* led to the discovery and characterization of three new spirostanol saponins, papolatiosides A–C (**1**–**3**), and nine known compounds (**4**–**12**). Their structures were established via extensive spectroscopic data analysis and chemical methods. Interestingly, compounds **1** and **2** possessed a fructosyl in their oligosaccharide moiety, which is rare in natural product and was firstly reported in family Melanthiaceae. The cytotoxicity of these saponins against several human cancer cell lines was evaluated by a CCK-8 experiment. As a result, compound **1** exhibited a significant cytotoxic effect on LN229, U251, Capan-2, HeLa, and HepG2 cancer cells with IC_50_ values of 4.18 ± 0.31, 3.85 ± 0.44, 3.26 ± 0.34, 3.30 ± 0.38 and 4.32 ± 0.51 μM, respectively. In addition, the result of flow cytometry analysis indicated that compound **1** could induce apoptosis of glioma cells LN229. The underlying mechanism was explored by network pharmacology and western bolt experiments, which indicated that compound **1** could induce glioma cells LN229 apoptosis by regulating the EGFR/PI3K/Akt/mTOR pathway.

## 1. Introduction

Natural products have contributed significantly to the discovery of novel anticancer drugs. Steroidal saponins, as common natural products, are found in a wide range of natural plants, including traditional Chinese medicine such as *Paris polyphylla* and *Panax ginseng*. A series of steroidal saponins with good antitumor or antiviral activity were also produced, including diosgenin, *Paris* saponin H, Ginsenoside Rd, and so on. Steroidal saponins have demonstrated outstanding pharmacological effects, such as anti-inflammatory [1], anti-viral [2], immunomodulatory [3], and antitumor [4] activities. Especially, modern pharmacological research has shown that steroidal saponins exhibited a significant resistant effect to different cancer cells through a variety of molecular mechanisms, such as apoptosis, cell cycle arrest, inhibition of migration, and invasion [5,6,7,8]. Therefore, steroidal saponins might be the source of new potential anti-cancer drugs.

The genus *Paris* (family Melanthiaceae) consists of 33 species and 15 varieties, most of which are distributed in the tropical and temperate regions of Asia and Europe [9]. It is a famous traditional Chinese medicine with antitumor, hemostatic, antimicrobial, and antalgic properties. The dried rhizomes of *Paris polyphylla* var. *chinensis* and *Paris polyphylla* var. *yunnanensis* are documented as Paridis Rhizoma to elucidate the medicinal value and quality standard of *Paris* in the 2020 edition of *Chinese Pharmacopoeia*. The phytochemical investigation revealed that steroidal saponins were the predominant secondary metabolites in genus *Paris*. *Paris polyphylla* var. *latifolia* [10], a variant of the genus *Paris*, is widely distributed in southwest China. To date, it has only been previously studied in our research group [11,12]. Therefore, this study focused on the cytotoxic steroidal saponins in *Paris polyphylla* var. *latifolia*. As a result, three new steroidal saponins (**1**–**3**) and nine known ones (**4**–**12**) were obtained from the rhizomes of *Paris polyphylla* var. *latifolia*. Compounds **1** and **2** possessed a fructose in their oligosaccharide moiety, which was rare in natural product and was firstly reported in the family Melanthiaceae. Then, the inhibitory effect of the isolated compounds against several human cancer cell lines was evaluated using the CCK-8 method. Compounds **1** and **2** exhibited a significant inhibition effect on cancer cells with IC_50_ values from 3.26 ± 0.34 μM to 22.3 ± 1.33 μM. Moreover, the underlying mechanism of *Paris* saponins inhibiting the proliferation of glioma cells was explored by network pharmacology, and the mechanism of compound **1** on the apoptosis of glioma cells was further confirmed by flow cytometry and western blot experiments. Herein, we report the isolation, structure elucidation, cytotoxic evaluation, and mechanism exploration of the steroidal saponins from *Paris polyphylla* var. *latifolia*.

## 2. Results

### 2.1. Isolated Compounds from Paris polyphylla var. latifolia

Separation of the water saturated *n*-BuOH fraction of an ethanol extract of the rhizomes of *Paris polyphylla* var. *latifolia* by silica gel chromatography and preparative HPLC afforded three new spirostanol saponins, papolatiosides A–C (**1**–**3**), along with nine known compounds (**4**–**12**) (Figure 1).

### 2.2. Structure Elucidation of the New Compounds

Compound **1**, white amorphous powder, was positive to Liebermann Burchard and Molisch chemical reactions, which indicated that it might be a steroidal glycoside. Its molecular formula was determined as C_45_H_72_O_17_ based on the ion peak of HR-ESI-MS at *m/z* 885.4854 [M + H] ^+^ (calculated for C_45_H_73_O_17_, 885.4848). In the ^1^H-NMR spectrum (Table 1), four methyl protons were detected at *δ*_H_ 0.80 (3H, s, H-18), 1.05 (3H, s, H-19), 0.96 (3H, d, *J* = 7.0 Hz, H-21), and 0.79 (3H, d, *J* = 6.5 Hz, H-27), and one olefinic methine proton signal was found at *δ*_H_ 5.38 (1H, br s, H-6). In the ^13^C-NMR spectrum (Table 1), the corresponding methyl carbon signals were observed at *δ*_C_ 16.8 (C-18), 19.9 (C-19), 14.9 (C-21), and 17.5 (C-27), and the trisubstituted double bond carbons were obtained at *δ*_C_ 141.89 (C-5) and *δ*_C_ 122.65 (C-6). Moreover, there was a quaternary ketal carbon at *δ*_C_ 110.6 (C-22). In the HMBC spectrum, the cross peaks between H-19 (*δ*_H_ 1.05) and C-5 (*δ*_C_ 141.89) and between H-6 (*δ*_H_ 5.38) and C-4 (*δ*_C_ 39.6)/C-8 (*δ*_C_ 32.8)/C-10 (*δ*_C_ 38.0) indicated that the double bound was located at C-5/C-6. Therefore, the aglycone was identified as diosgenin with a double bond located at C-5/C-6 by analyzing NMR data and comparing it with known compounds [13].

The ^13^C NMR spectrum showed 45 carbon signals, 27 of which were assigned to the aglycone moiety and the remaining 18 carbons to the sugar moiety. The presence of monosaccharides in **1** was further confirmed by acid hydrolysis and GC analyses. The results of the GC-MS analysis showed D-glucose (D-Glc), L-rhamnose (L-Rha), and D-fructose (D-Fru) in a ratio of 1:1:1 by comparing the retention times with the corresponding authentic samples. In the ^1^H-NMR spectrum, two anomeric protons were observed at *δ*_H_ 4.49 (d, *J* = 7.8 Hz, H-1 of Glc) and *δ*_H_ 5.20 (br s, H-1 of Rha), and the corresponding carbon signals were obtained at *δ*_C_ 100.9 (C-1 of Glc) and *δ*_C_ 102.2 (C-1 of Rha) by analyzing the HSQC correlations, respectively. In the ^13^C-NMR spectrum, there was a specific quaternary carbon at *δ*_C_ 105.2, which was identified as the anomeric carbon of D-Fru. Then, the assignments of protons and carbons in each monosaccharide were determined by analyzing the ^1^H-^1^H COSY, HSQC, HMBC and TOCSY spectra (Table 2). The *β* configuration of D-glucopyranosyl was revealed by the anomeric proton coupling constants (*J* = 7.8 > 7.0 Hz) and the *α* configuration of L-rhamnopyranosyl was confirmed by the chemical shifts of C-5 of Rha (*δ*_C_ 69.8) [14,15]. The *β* configuration of D-fructofuranosyl was confirmed by the chemical shifts of *δ*_C_ 62.1 (C-1 of Fru), 105.2 (C-2 of Fru), 79.2 (C-3 of Fru), 76.6 (C-4 of Fru), 83.7 (C-5 of Fru) and 64.2 (C-6 of Fru) [16]. The linkage sequences of monosaccharides were determined by the correlations in the HMBC spectrum, which were observed from *δ*_H_ 3.66 (H-1 of Fru) to *δ*_C_ 61.6 (C-6 of Glc), *δ*_H_ 5.20 (H-1 of Rha) to *δ*_C_ 78.9 (C-2 of Glc), and *δ*_H_ 4.49 (H-1 of Glc) to *δ*_C_ 79.6 (C-3 of the aglycone) (Figure 2). The *α* orientation of H-3 was deduced from the correlations of H_3_-19 (*δ*_H_ 1.05) to H_β_-1 (*δ*_H_ 1.10), and H_α_-1 (*δ*_H_ 1.88) to H-3 (*δ*_H_ 3.58) in the NOESY spectrum [15]. The 25*R* configuration was deduced from the difference in the geminal protons of H-26 (∆*δ*_H_ = 0.12 < 0.48) [17]. Thus, the structure of compound **1** was elucidated as diosgenin-3-*O*-*β*-D-fructofuranosyl-(1→6)-[*α*-L-rhamnopyranosyl-(1 → 2)]-*β*-D-glucopyranoside.

Compound **2**, white amorphous powder, was positive to Liebermann Burchard and Molisch chemical reactions, which indicated that it might be a steroidal glycoside. Its molecular formula was determined as C_50_H_80_O_21_, which was deduced from the ion peak of HR-ESI-MS at *m/z* 1017.5258 [M + H] ^+^ (calculated for C_50_H_81_O_21,_ 1017.5270). In the ^1^H-NMR spectrum, four methyl protons were detected at *δ*_H_ 0.81 (3H, s, H-18), 1.04 (3H, s, H-19), 0.96 (3H, d, *J* = 7.0 Hz, H-21), and 0.79 (3H, d, *J* = 6.5 Hz, H-27), and one olefinic methine proton signal was found at *δ*_H_ 5.38 (1H, br s, H-6) (Table 1). In the ^13^C-NMR spectrum, the corresponding methyl carbon signals were observed at *δ*_C_ 16.8 (C-18), 19.8 (C-19), 14.9 (C-21), and 17.5 (C-27) and the trisubstituted double bond carbons were obtained at *δ*_C_ 141.9 (C-5) and *δ*_C_ 122.7 (C-6). The comparison of the ^1^H-NMR and ^13^C-NMR signals of compound **2** with those of **1** indicated these two compounds possessed the same aglycone moiety. The difference was found in the oligosaccharide moiety.

The results of acid hydrolysis and GC-MS analysis revealed the presence of D-glucose (D-Glc), L-rhamnose (L-Rha), L-arabinose (L-Ara), and D-fructose (D-Fru) in a ratio of 1:1:1:1 by comparing the retention times with the corresponding authentic samples. In the ^1^H-NMR and ^13^C-NMR spectra, three anomeric proton signals were offered at *δ*_H_ 4.53 (d, *J* = 7.81 Hz, H-1 of Glc), 5.23 (br s, H-1 of Rha), and 5.11 (d, *J* = 1.63 Hz, H-1 of Ara) with the corresponding carbon signals at *δ*_C_ 100.5 (C-1 of Glc), *δ*_C_ 102.1 (C-1 of Rha), and *δ*_C_ 109.77 (C-1 of Ara). In the ^13^C-NMR spectrum, the characteristic quaternary carbon signal at *δ*_C_ 109.6 was identified as the anomeric carbon of D-Fru. The assignments of ^1^H-NMR and ^13^C-NMR signals associated with the oligosaccharide moiety were derived from the ^1^H-^1^H COSY, HSQC, HMBC, and TOCSY spectra (Table 2). In the HMBC spectrum, the cross peaks from H-1 (*δ*_H_ 5.11, L-Ara) to C-4 (*δ*_C_ 86.0, L-Ara) and H-4 (*δ*_H_ 4.07, L-Ara) to C-1 (*δ*_C_ 109.8, L-Ara) revealed the formation of L-arabinofuranosyl. The *β* configuration of D-glucopyranosyl and D-fructofuranosyl and the *α* configuration of L-rhamnopyranosyl were identified the same way with compound 1 (Figure 3). The small ^3^*J*_H-1, H-2_ (1.63 Hz) of H-1 of L-Ara*f* and the characteristic carbon signals of C-1 (*δ*_C_ 109.8), C-2 (*δ*_C_ 83.2), C-4 (*δ*_C_ 86.0), and C-5 (*δ*_C_ 62.9) suggested the *α* configuration of L-Ara*f* [18]. The linkage of monosaccharide was determined by the correlations from *δ*_H_ 5.23 (H-1 of Rha) to *δ*_C_ 78.5 (C-2 of Glc), *δ*_H_ 5.11 (H-1 of Ara*f*) to *δ*_C_ 78.7 (C-4 of Glc), *δ*_H_ 3.68 (H-1 of Fru) to *δ*_C_ 61.0 (C-6 of Glc), and *δ*_H_ 4.53 (H-1 of Glc) to *δ*_C_ 79.7 (C-3 of the aglycone) in the HMBC spectrum (Figure 3). Based on the above evidence, the structure of compound **2** was elucidated as diosgenin-3-*O*-*β*-D-fructofuranosyl- (1 → 6)- [*α*-L-arabinofuranosyl- (1→4)]- [*α*-L-rhamnopyranosyl- (1→2)]-*β*-D-glucopyranoside.

Compound 3 was obtained as white amorphous powder and was positive to Liebermann Burchard and Molisch chemical reactions. Its molecular formula was determined as C_39_H_58_O_16_ based on the ion peak of HR-ESI-MS at *m/z* 781.3653 [M − H] ^+^ (calculated for C_39_H_57_O_16_, 781.3647). In the ^1^H-NMR spectrum, two methyl protons were observed at *δ*_H_ 0.68 (3H, s, H-18), 1.05 (3H, s, H-19), and one olefinic methine proton signal was found at *δ*_H_ 5.39 (1H, br s, H-6) (Table 1). In the ^13^C-NMR spectrum, the corresponding methyl carbon signals were observed at *δ*_C_ 14.7 (C-18), 19.8 (C-19) and the trisubstituted double bond carbons were obtained at *δ*_C_ 142.0 (C-5) and *δ*_C_ 122.4 (C-6). Additionally, one carbonyl carbon signal was detected at *δ*_C_ 173.45 (C-22). By comparing the ^1^H-NMR and ^13^C-NMR data with those of the known compound 5 [19], the aglycone of 3 was determined to be the same as **5,** which contained a γ-butyrolactone structure in ring E.

Acid hydrolysis and GC-MS analysis of **3** revealed that the monosaccharides were D-Glc, L-Rha and L-Ara in a ratio of 1:1:1. This was further confirmed by the anomeric signals at *δ*_H_ 4.50 (d, *J* = 7.81 Hz, H-1 of Glc) with *δ*_C_ 100.4 (C-1 of Glc), *δ*_H_ 5.22 (br s, H-1 of Rha), with *δ*_C_ 102.1 (C-1 of Rha), and *δ*_H_ 5.01 (d, *J* = 1.68 Hz, H-1 of Ara) with *δ*_C_ 109.9 (C-1 of Ara) in the ^1^H NMR and ^13^C-NMR spectra (Table 2). The *β* configuration of D-glucopyranosyl and the *α* configurations of L-arabinopyranosyl and L-rhamnopyranosyl were identified as the same way with compounds **1** and **2**. The linkage sequence of monosaccharide was determined by the correlations in the HMBC spectrum, which were from *δ*_H_ 5.22 (H-1 of Rha) to *δ*_C_ 78.8 (C-2 of Glc), *δ*_H_ 5.01 (H-1 of Ara*f*) to *δ*_C_ 78.6 (C-4 of Glc), and *δ*_H_ 4.50 (H-1 of Glc) to *δ*_C_ 79.2 (C-3 of the aglycone) (Figure 4). Thus, the structure of **3** was determined to be 3*β*,16*β*-dihydroxy-pregn-5,20-dien-carboxylic acid (22,16)-lactone-3-*O*-*α*-L-arabinopyranosyl-(1 → 4)-*O*-[*α*-L-rhamnopyranosyl-(1→2)]-*β*-D-glucopyranoside.

### 2.3. Structure Identification of the Known Isolated Compounds

The nine known compounds were identified as pregna-5,16-diene-3*β*-ol-20-one-3-*O*-*α*-L-arabinofuranosyl-(1 → 4)-[*α*-L-rhamnopyranosyl(1 → 2)]-*β*-D-glucopyranoside (**4**) [20], chonglouoside SL-8 (**5**) [19], pallidifloside D (**6**) [21], parispseudoside A (**7**) [22], 3-*O*-*α*-L-arabinofuranosyl-(1 → 4)-[*α*-L-rhamnopyranosyl (1 → 2)]-*β*-D-glucopyranosyl-*β*-D-chacotriosyl-26-*O*-*β*-D-glucopyranoside (**8**) [23], aethioside A (**9**) [24], spongipregnolosides A (**10**) [25], hypoglaucin H (**11**) [26], parispseudoside B (**12**) [24] by using the same set of methods as the new compounds.

### 2.4. Cytotoxicities of Compounds ***1***–***12*** against Cancer Cell lines

The CCK-8 method was used to evaluate the cytotoxicity of the isolated compounds against the HeLa, U251, LN229, Capan-2, HepG2, and NHA cell lines. The results showed that compounds **1** and **2** exhibited significant cytotoxic activity, with IC_50_ values ranging from 3.26 ± 0.34 μM to 22.3 ± 1.33 μM against five cancer cell lines, while the remaining compounds were inactive. None of the compounds had significant cytotoxic activity against normal human NHA astrocytes (Table 3). The results above were consistent with the findings in our previous research, which indicated that spirostanol saponins bound with oligosaccharide at C-3 of the aglycone possess stronger cytotoxicity than the furostanol saponins and cholestanol-type saponins, which means that the opening of the F-ring could reduce the cytotoxic activity of spirostanol saponins [27]. Compound **1** is the first fructosyl-containing steroid saponins in Melanthiaceae, which has strong cytotoxicity. Moreover, the cytotoxicity of **1** and **2** confirmed that the spirostanol saponins possessing a three-glycosyl oligosaccharide chain at C-3 showed higher cytotoxicity than the others.

### 2.5. Prediction of Potential Targets of Paris Saponins against Glioma by Network Pharmacology

#### 2.5.1. The Collection of Potential Targets of *Paris* Saponins

According to the literature, seven common spirostanol *Paris* saponins with strong antitumor activity were selected. The 2D molecular structure of *Paris* saponins (polyphyllin Ⅰ, polyphyllin Ⅱ, polyphyllin Ⅵ, paris saponin Ⅶ, polyphyllin H, dioscin and gracillin) were obtained from the PubChem database, and the SMILES of *Paris* saponins were confirmed by CAS number in the Pubchem database. As a result, more than 100 targets for each *Paris* saponins were found. After summarizing and removing the repeated parts, a total of 207 targets were identified for these compounds (Figure 5).

#### 2.5.2. The Common Targets of *Paris* Saponins against Glioma

A total of 5435 glioma-related gene targets were found in the GeneCards database. The Glioma-related genes and drug-target genes were screened through Venn mapping to screen out 151 common targets. The Uniprot database was used to rectify and normalize (Figure 6).

#### 2.5.3. PPI Network of *Paris* Saponins and Glioma

PPI networks can study the molecular mechanisms and drug targets of complex diseases from a systematic perspective. Cytoscape was used to treat the targets. The darker the color in the figure, the deeper the interaction with other targets. A PPI network of *Paris* saponins and glioma-related targets was constructed (Figure 7). The circles represent proteins, and the lines represent interactions, and 151 targets were summarized (Figure 8). A total of 151 targets were involved and a total of 33 major targets were screened by MCODE. As a result, GRB2, SRC, STAT3, PIK3CA, PTPN11 have the most interactions in 33 circles (Figure 9).

#### 2.5.4. KEGG Function Analysis

KEGG analysis revealed that key targets were mainly enriched in 9 signal pathways (Figure 10), including EGFR tyrosine kinase inhibitor resistance, the Ras signaling pathway, the Calcium signaling pathway, Adherens junction, Gap junction, Transcriptional misregulation in cancer, bladder cancer, the Adipocytokine signaling pathway and Neuroactive ligand-receptor interaction. Among them, EGFR tyrosine kinase inhibitor resistance and the Ras signaling pathway are mostly enriched. The results suggest that *Paris* saponins may play a role in the treatment of glioma through the aforementioned ways.

#### 2.5.5. GO Function Enrichment Analysis

GO enrichment analysis showed the anti-tumor effect of *Paris* saponins in biological processes. The abscissa is the GO pathway, and the ordinate is the enrichment score. The results are shown in bubble charts. The results show that BP have positive regulation in the transmembrane receptor protein tyrosine kinase signaling pathway (Figure 11). The enrichment target in CC is mostly receptor complex (Figure 11). The enrichment targets in MF are mostly plasma membrane signaling receptor complex (Figure 11).

### 2.6. Compound ***1*** Induce the Apoptosis of Glioma Cells LN229 by Activating EGFR/PI3K/Akt/Mtor Pathway

Flow cytometry analysis was used to measure the apoptosis effect of compound **1** against glioma cells LN229. Annexin V^+^/PI^–^ represents the apoptotic cells and annexin V^−^/PI^−^ represents the living cells. After being treated with different concentrations of compound **1** for 24 h, the ratio of apoptotic LN229 was increased, while the ratio of living cells was decreased, which indicated that compound **1** could induce the apoptosis of glioma cells significantly (Figure 12a). Then, the expression of the apoptosis-related proteins was examined by western blot experiment. The results showed that the expression of Bax/Bcl-2, cleaved caspase 3, and cytochrome c in the compound **1** treated with LN229 were significantly increased compared with the control group (Figure 12b). Furthermore, the results of western blot showed that compound **1** significantly suppressed the phosphorylation levels of EGFR in LN229, while having no evident effect on the expression of total EGFR. Similarly, the expression level of p-PI3K, p-Akt, and p-mTOR was decreased in LN229 cells after treatment with compound **1** (Figure 12c). In conclusion, all these data suggested that compound **1** treatment could inactivate EGFR/PI3K/Akt/mTOR signaling in glioma cells LN229.

## 3. Discussion

According to the literature, D-glucose, L-arabinose, and D-rhamnose are common in the oligosaccharide of steroidal saponins, while fucose is rare [9]. Therefore, fructosyl has only been found in the natural product once from the family Agavaceae [16]. Interestingly, two naturally occurring fructosyl-containing steroidal saponins were isolated and identified from *P*. *polyphylla* var. *latifolia* in this study. Furthermore, this is the first study to report fructosyl-containing steroidal saponins from genus *Paris* and the family Melanthiaceae.

The method of network pharmacology was used to explore the underlying mechanism of compound **1** to inhibit the proliferation of glioma cells LN229. Network pharmacology is a drug research strategy proposed on the basis of system biology and multidirectional pharmacology. By constructing the “glioma-*Paris* saponins” interaction network and screening key targets and biological processes, we evaluated the potential mechanism of *Paris* saponins against glioma from the overall level, and tested the accuracy of the network pharmacological prediction by taking the new compound **1** as an example in vitro.

EGFR is overexpressed in most human glioblastomas and plays a key role in tumor formation [28]. Therefore, treatments specifically targeting EGFR and its downstream pathways in glioblastoma cells may have potential therapeutic effects. It is well known that the PI3K/Akt/mTOR signaling pathway is regulated by EGFR and is closely related to cell growth, autophagy, and proliferation [29]. In this study, key proteins in the EGFR/PI3K/Akt/mTOR signaling pathway, such as p-EGFR, p-PI3K, p-Akt, and p-mTOR, were significantly inhibited by compound **1**.

In summary, the phytochemical investigation of *Paris polyphylla* var. *latifolia* resulted in the isolation of three new saponins (**1**–**3**), and nine known saponins (**4**–**12**). Compounds **1** and **2** possessed a fructosyl in their oligosaccharide moiety, which was rare in natural product and was firstly reported in family Melanthiaceae. Meanwhile, the results of cytotoxic assay showed that compounds **1** and **2** possessed significant cytotoxic effects on LN229, U251, Capan-2, HeLa and HepG2 cancer cells. The result of network pharmacology screening indicated that EGFR might be the potential target of Paris saponins against glioma cells. In addition, the results of flow cytometry and western bolt experiments confirmed that compound **1** induces glioma cell apoptosis by regulating the EGFR/PI3K/Akt/mTOR pathway. The findings in this study indicated that compound **1** could be a potential inhibitor of EGFR for the treatment of glioma.

## 4. Materials and Methods

### 4.1. General Experimental Procedures

Optical rotations were measured on a Perkin-Elmer 241 MC digital polarimeter (German PerkinElmer Corporation, Boelingen, Germany). 1D and 2D-NMR spectral experiments were measured in CD_3_OD on a Bruker AVANCE-800 spectrometer (Bruker Corporation, Karlsruhe, Germany) with TMS as an internal standard. The HR-ESI-MS spectra were carried out on AB SCIEX X500R high-resolution liquid chromatography-mass spectrometer (SCIEX, Framingham, MA, USA). GC-MS analysis was performed on an Agilent GC-MS 6890-5973 instrument equipped with an RXI-5 SIL MS column (30 m × 0.25 mm × 0.25 µm). Column chromatography (CC) was operated on a Sephadex LH-20 (GE-Healthcare, Uppsala, Sweden), Pure C-810 middle pressure liquid chromatography (BUCHI, Flawil, Switzerland), and silica gel H (10−40 µm, Qingdao Marine Chemical Inc., Qingdao, China). Standards for D-glucose (D-Glc), L-rhamnose (L-Rha), D-fructose (D-Fru), and L-arabinose (L-Ara) were purchased from Sigma Chemical Co. (St. Louis, MO, USA).

### 4.2. Plant Materials

The rhizomes of *Paris polyohylla* var. *latifolia* were collected from Baoji, Shannxi Province, China, in August 2021 and identified by the corresponding author Hai-Feng Tang. The voucher specimen (No. 20210802) was deposited in the Department of Chinese Materia Medica and Natural Medicines, School of Pharmacy, Air Force Medical University, Xi’an, China.

### 4.3. Extraction and Isolation

The dried rhizomes of *Paris polyohylla* var. *latifolia* (2.8 kg) were chopped and refluxed with 70% ethanol (10.0 L) quintic (each 2 h). The ethanol solution was mixed and condensed by a vacuum rotary evaporator to receive a syrupy residue (611.7 g). Then, the syrupy residue was suspended in water (3.5 L) and extracted with petroleum ether and water saturated n-BuOH, successively. The water saturated n-BuOH layer was evaporated to afford the total saponins (135.0 g). The total saponins were separated by silica gel column chromatography and gradually eluted by CH_2_Cl_2_-MeOH-H_2_O (100:0:0, 50:1:0, 20:1:0, 8:1:0.1, 6:1:0.1, 7:2.5:0.1 and 6.5:3.5:0.1) to offer 19 sub-fractions (Fr.1–Fr.19) based on the result of the TLC analysis.

Fr.14 was purified by a Sephadex LH-20 column chromatography to remove pigmentum and to afford Fr.14-1 (4.2 g), Fr.14-2 (5.0 g), and Fr.14-3 (430.5 mg). Fr.14-1 was purified by a semi-preparative HPLC using MeCN-H_2_O (35:65) as the mobile phase at a flow rate of 15.0 mL/min to afford compounds **9** (5.7 mg, *t*_R_ = 28.9 min) and **10** (7.6 mg, *t*_R_ = 31.2 min).

Fr.15 was subjected to a Sephadex LH-20 column chromatography to remove pigmentum and was further purified by middle pressure liquid chromatography to obtain Fr.15-1 (128.5 mg) and Fr.15-2 (245.6 mg). Compounds **1** (9.8 mg, *t*_R_ = 18.0 min), **2** (15.8 mg, *t*_R_ = 18.5 min), **3** (15.2 mg, *t*_R_ = 21.0 min), **4** (25.6 mg, *t*_R_ = 21.8 min), **5** (28.5 mg, *t*_R_ = 25.0 min), **6** (40.7 mg, *t*_R_ = 29.0 min), and **11** (70.4 mg, *t*_R_ = 31.0 min) were offered by semi-preparative HPLC eluting with MeCN-H_2_O (40:60) at a flow rate of 15.0 mL/min from Fr.15-1.

Fr.16 was separated by silica gel column chromatography and eluted with CH_2_Cl_2_-MeOH-H_2_O (8:1:0.1, 8:2:0.2, 7:2.5:0.1, and 6:3:0.1) to give Fr.16-1 (1.1 g) and Fr.16-2 (430.5 mg). Fr.16-1 was separated on middle pressure liquid chromatography to afford Fr.16-1-1 (64.7 mg), Fr.16-1-2 (57.4 mg), and Fr-16-1-3 (145.3 mg). Then, Fr.16-1-1 and Fr.16-1-3 were isolated by semi-preparative HPLC eluting with MeCN-H_2_O (7:12, 2:3) at a flow rate of 15.0 mL/min to afford compounds **7** (9.1 mg, *t*_R_ = 24.3 min), **8** (21.4 mg, *t*_R_ = 28.5 min) and **12** (16.5 mg, *t*_R_ = 37.5 min), respectively.

Compound **1**: white amorphous solid; ${\left[\alpha \right]}_{D}^{22}=-71.2$ (*c* 0.1, MeOH); ^1^H (800 MHz, CD_3_OD) and ^13^C (200 MHz, CD_3_OD) NMR data, see Table 1 and Table 2; HR-ESI-MS *m/z* 885.4854 [M + H]^+^ (calculated for C_45_H_73_O_17_, 885.4848).Compound **2**: white amorphous solid; ${\left[\alpha \right]}_{D}^{22}=-87.3$ (*c* 0.1, MeOH); ^1^H (800 MHz, CD_3_OD) and ^13^C (200 MHz, CD_3_OD) NMR data, see Table 1 and Table 2; HR-ESI-MS *m/z* 1017.5258 [M + H]^+^ (calculated for C_50_H_81_O_21_, 1017.5270).Compound **3**: white amorphous solid; ${\left[\alpha \right]}_{D}^{22}=-129.5$ (*c* 0.1, MeOH); ^1^H (800 MHz, CD_3_OD) and ^13^C (200 MHz, CD_3_OD) NMR data, see Table 1 and Table 2; HR-ESI-MS *m/z* 781.3653 [M − H]^−^ (calculated for C_39_H_57_O_16,_ 781.3647).

### 4.4. Acid Hydrolysis of Compounds ***1***–***3*** and GC-MS Analysis

The absolute configuration of monosaccharides of compounds **1**–**3** was analyzed by GC-MS according to the method described in the literature [30]. Compounds **1**–**3** (2.0 mg each) were heated for 90 min at 110 °C in a tube containing 1 mL of 2 M trifluoroacetic acid (TFA). This dilutes the reaction mixture in 2 mL of distilled water and reduces it with 100 mg NaBH4, acetylated with acetic anhydride at 100 °C for 1 h. Standard monosaccharides (D-Glc, L-Rha, L-Ara*f* and D-Fru, 2.0 mg each) were treated as acetylated derivatives using the same methodology. Finally, the acetylated products were analyzed by the following methods: GC-MS was performed under the following conditions: carrier gas He (1.0 mL/min), injection volume of 1.0 μL, injector, temperature of 250 °C. The column temperature program rose within 120–250 °C at a rate of 3 °C/min and remained at 250 °C for 5 min.

The absolute configuration of the sugar fraction in compounds **1**–**3** was identified as D-Glc (*t*_R_ = 29.83 min), L-Rha (*t*_R_ = 18.62 min), L-Ara*f* (*t*_R_ = 19.94 min), and D-Fru (*t*_R_ = 30.87 min), comparing retention time to standard monosaccharides.

### 4.5. Network Pharmacologic Prediction

#### 4.5.1. The Collection of Potential Targets of *Paris* Saponins

The SMILES of *Paris* saponins (polyphyllin I, polyphyllin II, polyphyllin VI, paris saponin VII, polyphyllin H, dioscin and gracillin) were obtained from the PubChem database (https://pubchem.ncbi.nlm.nih.gov/) (accessed on 1 July 2022) by confirming the CAS number. The SMILES were imported into the Swisstarget database (http://www.swisstargetprediction.ch/) (accessed on 2 July 2022), the parameters were set, “Human Protein Targets Only” were set, and the targets were obtained. Next, the gene information and protein targets were corrected using the Uniprot database (http://www.uniprot.org/uploadlists/) (accessed on 3 July 2022) to obtain more reliable targets.

#### 4.5.2. The Collection of Gene Targets Related to Glioma

The GeneCards website (https://www.genecards.org/) (accessed on 23 June 2022) is a searchable, integrative database that provides comprehensive information on all annotated and predicted human genes. The knowledgebase automatically integrates gene-centric data from 150 web sources, including genomic, transcriptomic, proteomic, genetic, clinical, and functional information. The gene targets related to glioma were searched using this database, with “glioma” as the key word, to obtain gene targets related to glioma.

#### 4.5.3. Screening of Potential Targets

The Glioma-related targets and the targets of *Paris* saponins were screened using Venn (https://bioinfogp.cnb.csic.es/tools/venny/) (accessed on 6 July 2022). Their intersection could be the potential targets of *Paris* saponins against glioma.

#### 4.5.4. Construction of a Protein-Protein Interaction (PPI) Network

The String database (https://string-db.org/) (accessed on 7 July 2022) can be used as an effective protein interaction network database. Importing the potential targets to the String, the filter condition was set to “Human sapiens” and credibility to “medium confidence (0.400)”, then a Protein-Protein Interaction Network was built and the results were used to construct the primary target network in Cytoscape 3.9.0.

#### 4.5.5. Biological Function Annotation and Pathway Analysis

A total of 38 targets were enriched by KEGG and GO. We selected the top ten items in GO and constructed a bar chart. In addition, the significantly enriched pathways were screened and bubble graphs were drawn.

### 4.6. In Vitro Biological Experiments

#### 4.6.1. Cell Culture

U251 (human glioma cells), HeLa (human cervical cancer cells), and HepG2 (human hepatic cancer cells) were purchased from the Chinese Academy of Sciences (China), and Capan-2 (human pancreatic cancer cells) were obtained from Nanjing Cobioer Biosicences (Nanjing, China). American Type Culture Collection (ATCC) provided LN229 (human glioma cells). NHA (Gibco Human Astrocytes) and Gibco Astrocyte Medium (DMEM, N-2 Supplement, One Shot fetal bovine serum) were purchased from ThermoFisher Scientific (Waltham, MA, USA). The LN229, U251, HepG2 and HeLa cells were cultured in DMEM (Procell, Wuhan, China). The Capan-2 cells were cultured in RPMI-1640 (Procell, Wuhan, China) and NHA cells were cultured in Gibco Astrocyte Medium, supplemented with 10% FBS (Solarbio, Beijing, China) and 1% Penicillin–Streptomycin (Elabscience Biotechnology Co.,Ltd, Wuhan, China) at 37 °C with 5% CO_2_.

#### 4.6.2. Antibodies and Reagents

Primary antibodies against Bax (50599-2-Ig), Bcl-2 (26593-1-AP), Akt (10176-2-AP), p-Akt (28731-1-AP), PI3K (20584-1-AP), p-PI3K (60225-1-Ig), mTOR (28273-1-AP), p-mTOR (67778-1-Ig), Cytochrome C (10993-1-AP), β-actin (20536-1-AP), HRP-Goat-Anti-Mouse IgG (H+L) (SA00001-1), and HRP-Goat-Anti-Rabbit IgG (H+L) (SA00001-2) were obtained from Proteintech Group, Inc. (Rosemont, USA). Primary antibodies against EGFR (WL0682a), p-EGFR (WL03432) and cleaved caspase 3 (WL01992) were obtained from Wanleibio (Shenyang, China).

#### 4.6.3. Cytotoxic Assay

In the exponential phase of the growth, five tumor cell lines and one normal human astrocytes were plated onto 96-well plates at a concentration of 8000 cells/well for 24 h. Compounds **1**–**12** were prepared to various concentrations (64, 32, 16, 8, 4, 2, 1 and 0.5 µM in medium containing less than 0.1% DMSO) and incubated in 96-well plates (each concentration in four-fold wells) for 24 h. Cell viability was determined according to reported assay methods using the commercial CCK-8 kit (Elabscience Biotechnology Co., Ltd., Wuhan, China). The optical density (OD) of each well was measured with an AMR-100 microplate reader at 450 nm (Allsheng Corporation, Hangzhou, China). Cytotoxicity emerged as the value of the drug concentration at the inhibition of cell growth by 50% (IC_50_). The experiments were conducted for three independent replicates, and Doxorubicin was used as the positive control.

#### 4.6.4. Flow Cytometry Analysis for Apoptosis

Flow cytometry was used to analyze the effect of **1** on the apoptosis of glioma cells LN229. Cells were treated with different concentrations of **1** for 24 h. Then, the cells were collected and washed twice with cold PBS for five minutes (1000 rpm). The cells were analyzed for apoptosis by flow cytometry (BD, Franklin Lakes, NJ, USA) using the Annexin V-FITC/PI assay kit.

#### 4.6.5. Western Blot Analysis

The LN229 cells were treated with various concentrations of **1** or 0.1% DMSO for 24 h. A BCA Protein Assay Kit was used to measure the protein concentration, then equal amounts of protein samples were separated by 10% sodium dodecyl sulfate–polyacrylamide gel electrophoresis and transferred to a polyvinylidene fluoride membrane (Millipore, Burlington, MA, USA). The membranes were blocked for 2 h in Tris-buffered saline with PBST buffer, containing 5% skim milk. Then, the membranes were incubated with the appropriate primary antibodies overnight at 4 ℃. Then, the membranes were washed with PBST and reacted with a secondary horseradish peroxidase-conjugated antibody. Image Lab 5.2.1 (Biorad Laboratories, Hercules, CA, USA) was used to analyze the blots after detection using enhanced chemiluminescence (Millipore). Data analyses were performed using the GraphPad Prism software (San Diego, CA, USA).

### 4.7. Statistical Analysis

Data from three independent experiments were presented as mean ± SD. Differences between various experimental and control groups were compared using one-way analysis of variance followed by unpaired Student’s T test, and *p*-values < 0.05 were considered statistically significant.

## Figures and Tables

**Figure 1 ijms-24-07149-f001:**
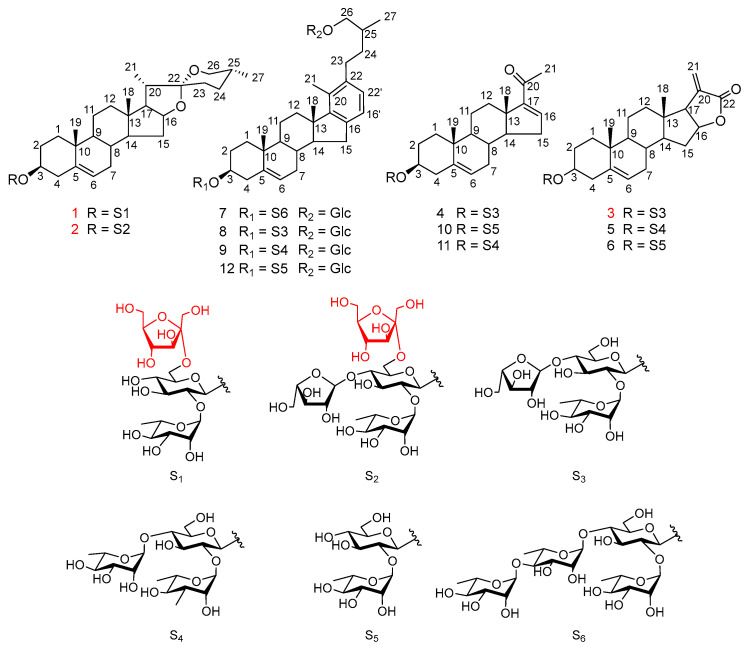
Chemical structures of compounds **1**–**12**.

**Figure 2 ijms-24-07149-f002:**
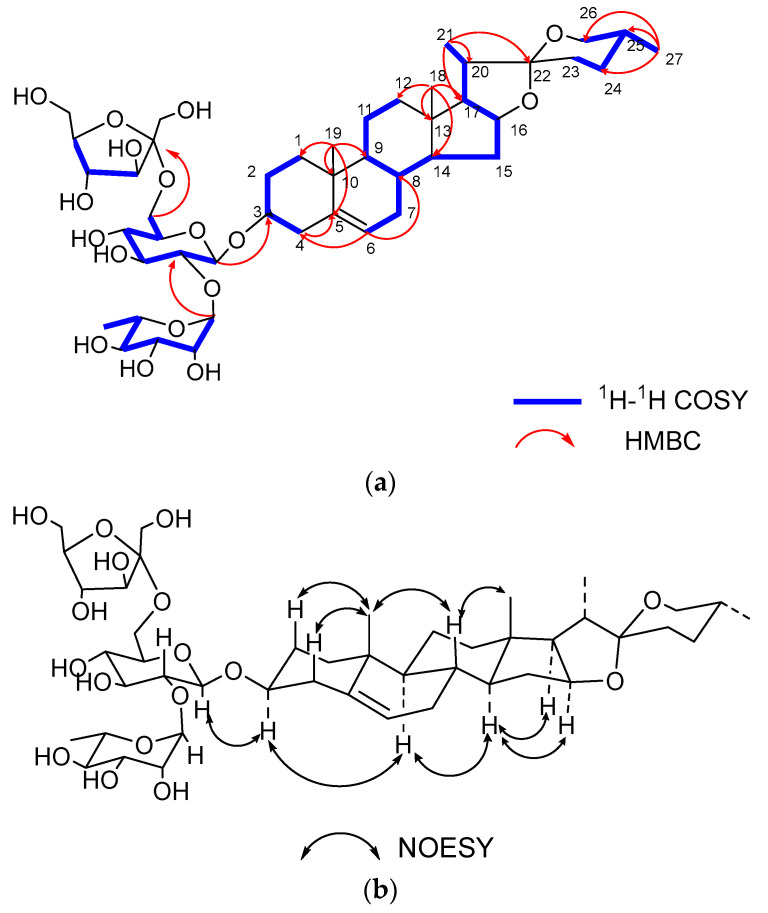
(**a**) Key ^1^H-^1^H COSY, HMBC correlations of compound **1**. (**b**) Key NOESY correlations of compound **1**.

**Figure 3 ijms-24-07149-f003:**
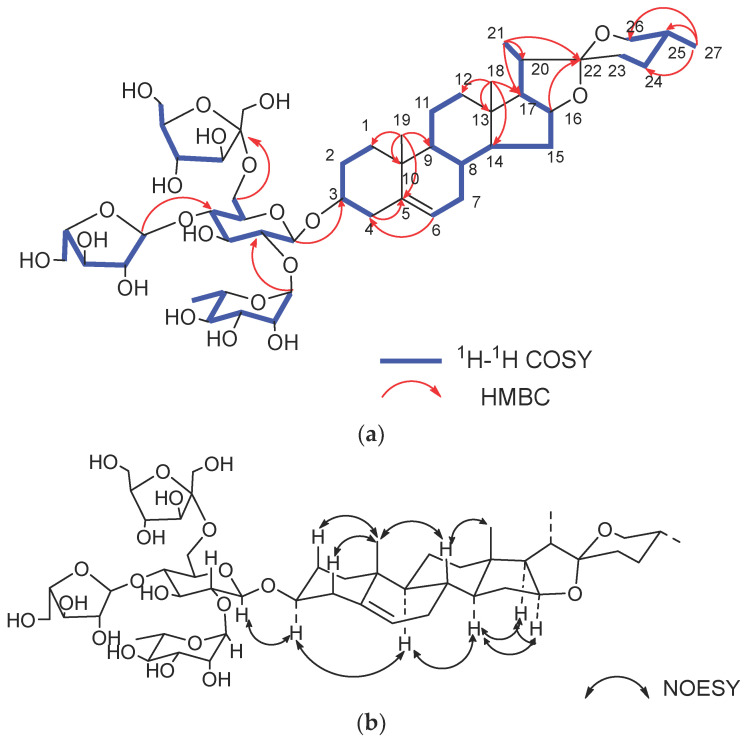
(**a**) Key ^1^H-^1^H COSY and HMBC correlations of compound **2**. (**b**) Key NOESY correlations of compound **2**.

**Figure 4 ijms-24-07149-f004:**
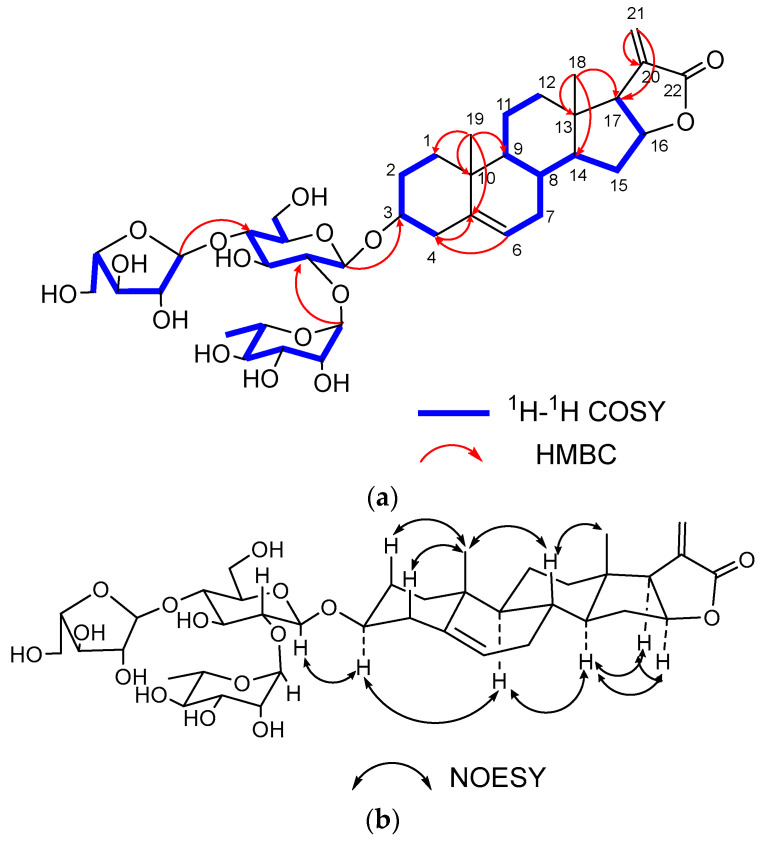
(**a**) Key ^1^H-^1^H COSY, HMBC correlations of compound **3**; (**b**) Key NOESY correlations of compound **3**.

**Figure 5 ijms-24-07149-f005:**
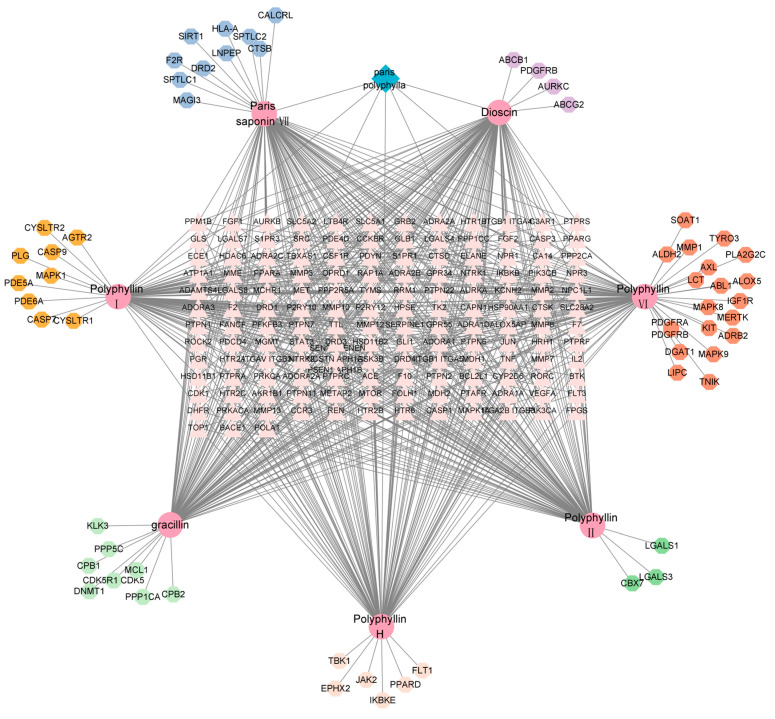
Intersectional targets of *Paris* saponins.

**Figure 6 ijms-24-07149-f006:**
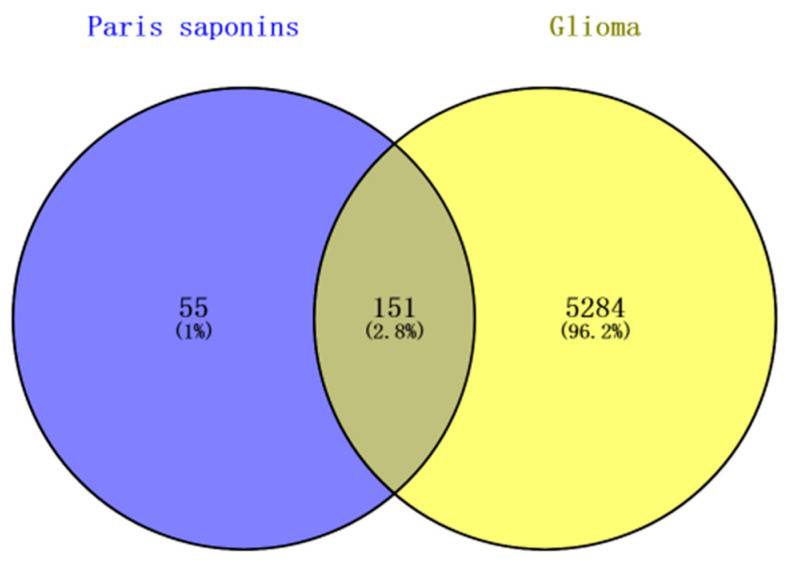
The Venn diagram of *Paris* saponins and glioma targets.

**Figure 7 ijms-24-07149-f007:**
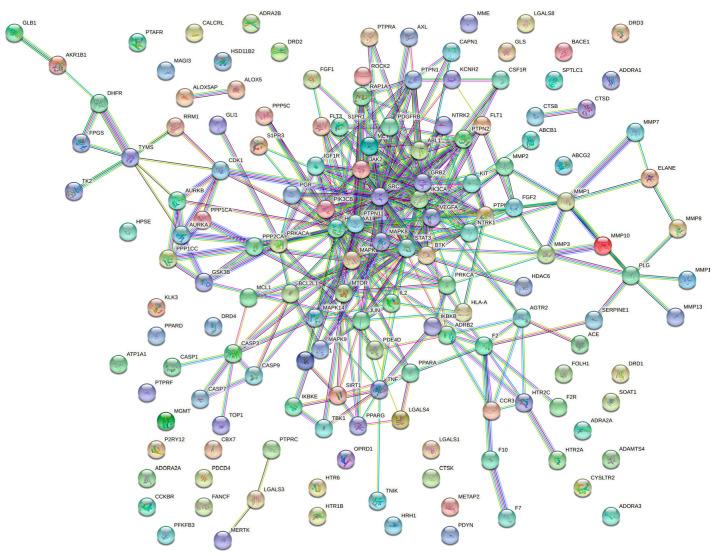
PPI network of the 151 targets.

**Figure 8 ijms-24-07149-f008:**
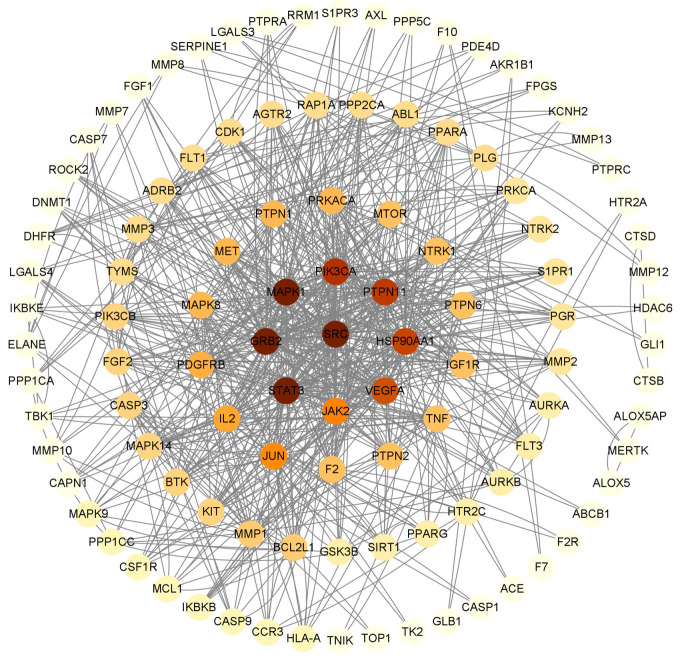
The target of *Paris* saponins in preventing and treating glioma cancer.

**Figure 9 ijms-24-07149-f009:**
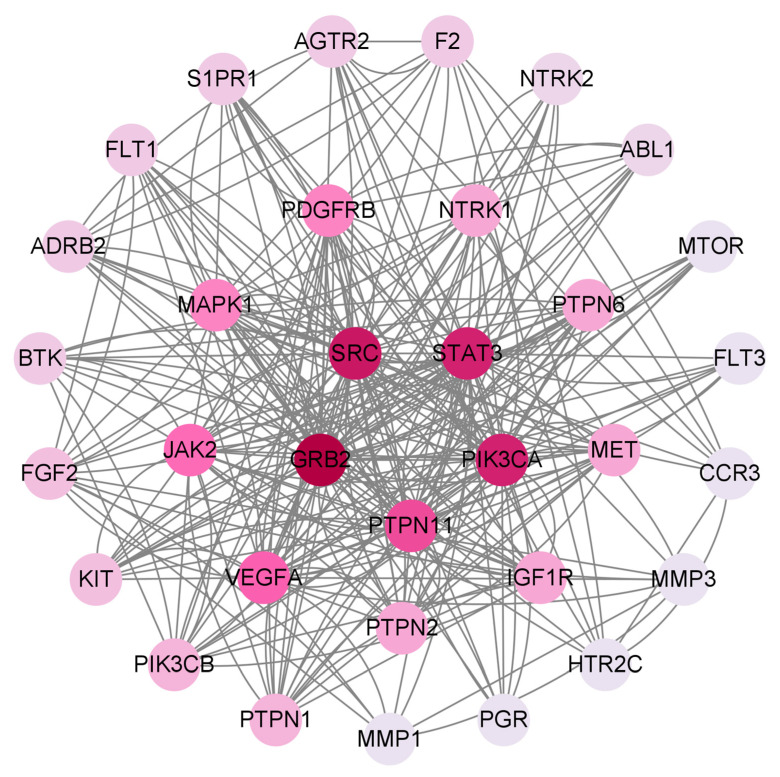
The key target of *Paris* saponins in preventing and treating glioma cancer screened by MCODE.

**Figure 10 ijms-24-07149-f010:**
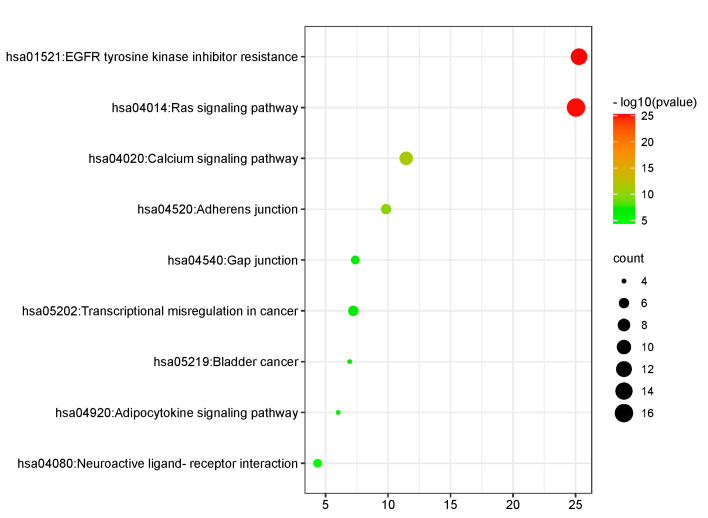
Analysis of KEGG Function.

**Figure 11 ijms-24-07149-f011:**
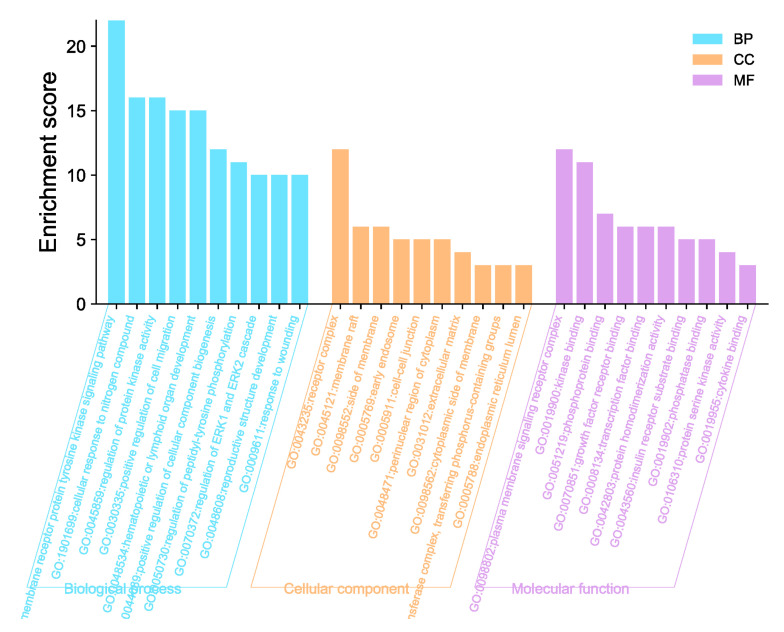
Analysis of GO Function.

**Figure 12 ijms-24-07149-f012:**
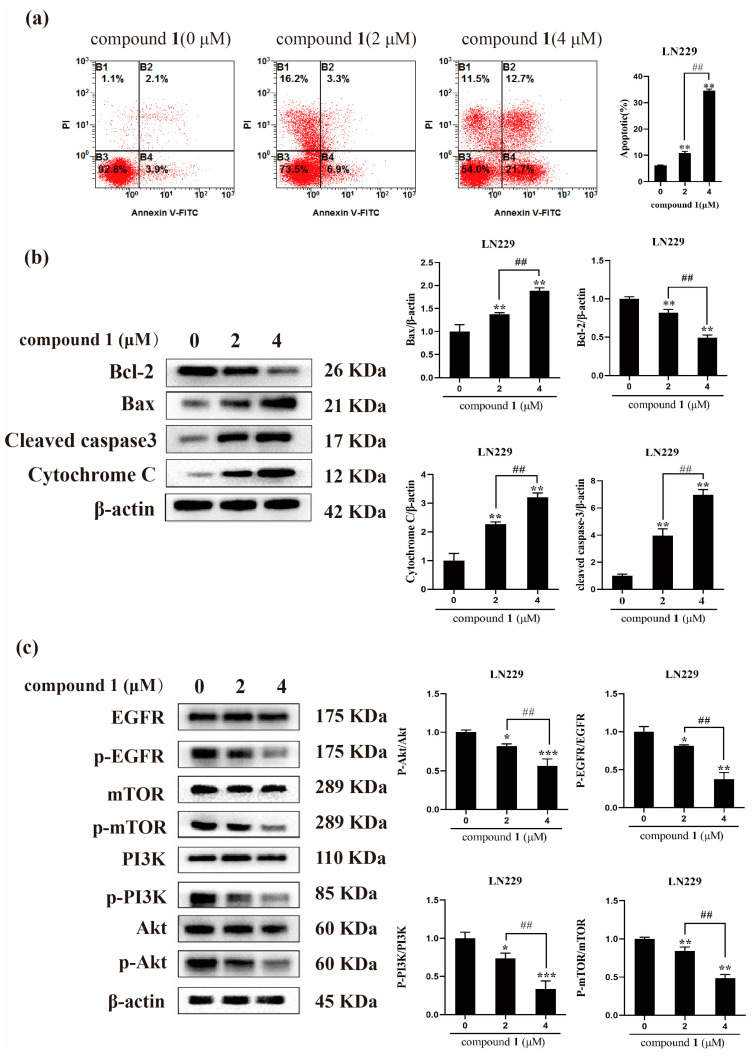
Compound **1** induced apoptosis of glioma cells LN229. (**a**) LN229 were treated with graded concentrations of compound **1** and stained with PI and Annexin V-FITC for flow cytometry analyses; (**b**) compound **1** increase the expression Bax/Bcl-2, cytochrome c and cleaved caspase 3; (**c**) compound **1** could regulated protein expression in EGFR/PI3K/Akt/mTOR signaling pathway. (n = 3, * *p* < 0.05, ** *p* < 0.01, *** *p* < 0.001, compared to the control group. ^##^
*p* < 0.01, compared to group 2 μM).

**Table 1 ijms-24-07149-t001:** ^13^C and ^1^H NMR data of aglycone moieties for compounds **1**–**3** in CD_3_OD.

Numbers	Compounds [*δ*_H_ mult. (*J* in Hz)]					
	1 ^a^		2 ^a^		3 ^a^	
1	38.5	1.88 o, 1.10 br s	38.5	1.86 o,1.10 br s	38.5	1.89 m, 1.09 br s
2	30.8	1.94 o, 1.29 o	30.7	1.94 o,1.29 o	30.7	1.92 m,1.29 o
3	79.6	3.58 m	79.7	3.54 m	79.2	3.59 m
4	39.6	2.44 m, 2.31 t	39.6	2.43 m, 2.30 t	39.5	2.45 m, 2.30 m
5	141.9	–	141.9	-	141.9	–
6	122.7	5.38 br s	122.7	5.38 br s	122.4	5.39 br s
7	33.2	2.01 m, 1.56 m	33.2	2.00 m, 1.56 m	33.0	2.04 m, 1.60 m
8	32.8	1.65 m	32.8	1.65 m	32.8	1.65 m
9	51.7	0.97 m	51.7	0.97 m	51.6	1.05 m
10	38.0	–	38.0	-	38.0	–
11	22.0	1.56 m, 1.51 br s	22.0	1.56 m, 1.50 br s	21.5	1.64 m, 1.54 m
12	40.9	1.76 o, 1.19 br s	40.9	1.76 m, 1.19 m	39.1	1.92 m, 1.36 br s
13	41.4	–	41.4	-	44.9	–
14	57.8	1.14 m	57.8	1.14 m	55.9	1.28 o
15	32.7	1.97 m, 1.28 o	32.7	1.98 m, 1.28 o	34.1	2.34 m, 1.54 m
16	82.2	4.39 q (7.4)	82.2	4.39 q (7.4)	83.7	4.91 m
17	63.7	1.74 br s	63.7	1.74 br s	56.3	2.93 dt (8.0, 1.8)
18	16.8	0.80 s	16.8	0.81 s	14.7	0.68 s
19	19.9	1.05 s	19.8	1.04 s	19.8	1.05 s
20	42.9	1.90 m	42.9	1.90 m	138.5	–
21	14.9	0.96 d (7.0)	14.9	0.96 d (7.0)	123.0	5.66 br s, 6.21 br s
22	110.6	–	110.6	-	173.5	–
23	32.4	1.70 d (4.6)	32.4	1.70 d (4.6)		
24	29.9	1.62 m, 1.41 m	29.9	1.62 m, 1.41 m		
25	31.4	1.59 o	31.5	1.59 m		
26	67.9	3.44 m, 3.32 o	67.9	3.44 m, 3.31 m		
27	17.5	0.79 d (6.5)	17.5	0.79 d (6.5)		

^a^ Tested in ^13^C-NMR (200 Hz) and ^1^H-NMR (800 Hz). o indicates overlapped.

**Table 2 ijms-24-07149-t002:** ^13^C and ^1^H NMR data of sugar portion for compounds **1**–**3** in CD_3_OD.

Sugars	1 ^a^		2 ^a^		3 ^a^	
	*δ* _C_	*δ*_H_ (*J* in Hz)	*δ* _C_	*δ*_H_ (*J* in Hz)	*δ* _C_	*δ*_H_ (*J* in Hz)
	Glc		Glc		Glc	
1	100.9	4.49 d (7.8)	100.5	4.53 d (7.8)	100.4	4.50 d (7.8)
2	78.9	3.38 m	78.5	3.39 m	78.8	3.40 m
3	79.1	4.08 m	77.7	3.62 m	77.8	3.25 m
4	71.2	3.42 m	78.7	3.87 m	78.6	3.51 m
5	76.2	3.32 m	74.8	3.51 m	76.5	3.35 m
6	61.6	3.93 m, 3.74 m	61.0	3.86 m, 3.79 m	61.9	3.82 m, 3.69 m
	Rha		Rha		Rha	
1	102.2	5.20 br s	102.1	5.23 br s	102.1	5.22 br s
2	72.2	3.90 m	72.2	3.88 m	72.2	3.89 m
3	72.4	3.65 m	72.4	3.65 m	72.4	3.65 m
4	74.0	3.39 m	73.9	3.38 m	73.9	3.39 m
5	69.8	4.12 m	69.8	4.12 m	69.7	4.12 m
6	18.0	1.23 d (6.3)	17.9	1.23 d (6.3)	17.9	1.23 d (6.3)
	Fru		Ara*f*		Ara*f*	
1	62.1	3.66 s, 3.57 s	109.8	5.11 d (1.6)	109.9	5.01 d (1.7)
2	105.2	-	83.2	3.99 m	83.2	3.97 m
3	79.2	3.46 d (8.7)	78.1	3.86 m	78.0	3.86 m, 3.62 m
4	76.6	4.01 t	86.0	4.07 m	85.9	4.07 m
5	83.7	3.73 d (4.9)	62.9	3.62 m	63.0	3.85 m, 3.72 m
6	64.2	3.72 d (9.9), 3.66 m				
			Fru			
1			60.5	3.70 s, 3.68 s		
2			109.6	-		
3			82.7	4.06 m		
4			79.0	3.57 t		
5			85.1	3.94 m		
6			63.0	3.62 m		

^a^ Tested in ^13^C-NMR (200 Hz) and ^1^H-NMR (800 Hz).

**Table 3 ijms-24-07149-t003:** Cytotoxicites of compounds **1**–**12** against five cancer cell lines.

Compounds	IC_50_ ± SD (μM)					
	LN229	U251	Capan-2	HeLa	HepG2	NHA
**1**	4.18 ± 0.31	3.85 ± 0.44	3.26 ± 0.34	3.30 ± 0.38	4.32 ± 0.51	>50
**2**	15.39 ± 0.47	15.76 ± 0.65	20.37 ± 1.05	22.3 ± 1.33	18.76 ± 1.25	>50
**3**	>50	>50	>50	>50	>50	>50
**4**	>50	>50	>50	>50	>50	>50
**5**	>50	>50	>50	>50	>50	>50
**6**	>50	>50	>50	>50	>50	>50
**7**	>50	>50	>50	>50	>50	>50
**8**	>50	>50	>50	>50	>50	>50
**9**	>50	>50	>50	>50	>50	>50
**10**	>50	>50	>50	>50	>50	>50
**11**	>50	>50	>50	>50	>50	>50
**12**	>50	>50	>50	>50	>50	>50
**Doxorubicin ^a^**	0.35 ± 0.05	0.21 ± 0.04	0.18 ± 0.02	0.18 ± 0.02	0.21 ± 0.02	>50

**^a^** Positive control.

## Data Availability

The data that support the findings of this study are available from the corresponding author upon reasonable request.

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
