# Peer review of "Cytotoxic Steroidal Saponins Containing a Rare Fructosyl from the Rhizomes of Paris polyphylla var. latifolia"

_ijms, 2023, doi:10.3390/ijms24087149_

Round 1

Reviewer 1 Report

This work is devoted to the isolation, identification and study of the biological properties of a number of compounds included in the rhizomes of Paris polyohylla var. latifolia. Three new compounds were identified and described. The structure of these compounds has been fully characterized using various methods of analysis, and the results is not in doubt. All obtained compounds were tested for cytotoxicity against five cancer cell lines, and two promising compounds with high micromolar activity were found. In addition, the result of network pharmacology screening indicated the potential target of Paris saponins against glioma cells. Also, the results of flow cytometry and western bolt experiments confirmed that compound 1 inducing glioma cell apoptosis by regulating EGFR/PI3K/Akt/mTOR pathway.

In my opinion, this is a very good and interesting work, including a detailed proof of the structure of new compounds, and also has the potential for further detailed study of biological properties.

This work will undoubtedly be of interest to the readers of the IJMS journal.

However, it is recommended to make the following corrections and additions before publication:

1) I would like to draw your attention to the fact that the work mentions an article of 2018 [reference 11], about identification steroidal saponins and lignan glycosides from the rhizomes of Paris polyphylla var. latifolia. But it doesn't mention the article «New Steroidal Saponins Isolated from the Rhizomes of Paris mairei» https://doi.org/10.3390/molecules26216366, in which compounds very similar in structure are described. I think it is necessary to mention all the literature where related compounds have been investigated.

2) Line 180. 2.3. Structure Identification of the Known Isolated Compounds.

It is not clear how compounds were identified. Using the same set of methods as for the new compounds? If using the full set of methods, then it is necessary to clarify in the text. "…characterized using the same set of methods as new compounds" for example.

3) In my opinion, when studying cytotoxic properties, data on a normal cell line should be added for control.

4) Unfortunately figure 5 is completely unreadable. Might be worth splitting it into separate parts. And also, higher quality of images is needed.

5) I also want to advise the authors to carefully review the entire text of the article and correct minor typos.

Author Response

Point 1: I would like to draw your attention to the fact that the work mentions an article of 2018 [reference 11], about identification steroidal saponins and lignan glycosides from the rhizomes of Paris polyphylla var. latifolia. But it doesn't mention the article «New Steroidal Saponins Isolated from the Rhizomes of Paris mairei» https://doi.org/10.3390/molecules26216366, in which compounds very similar in structure are described. I think it is necessary to mention all the literature where related compounds have been investigated. 

Response 1: Thank you for kind comment. The article you mentioned has been cited as reference 12 in the revised manuscript.

Point 2: Line 180. 2.3. Structure Identification of the Known Isolated Compounds.

It is not clear how compounds were identified. Using the same set of methods as for the new compounds? If using the full set of methods, then it is necessary to clarify in the text. "…characterized using the same set of methods as new compounds" for example.

Response 2: We are grateful for the suggestion. According to your advice, this paragraph has been corrected as “The nine known compounds were identified as pregna-5,16-diene-3β-ol-20-one-3-O-α-L-arabinofuranosyl-(1→4)-[α-L-rhamnopyranosyl(1→2)]-β-D-glucopyranoside (4) [20], chonglouoside SL-8 (5) [19], pallidifloside D (6) [21], parispseudoside A (7) [22], 3-O-α-L-arabinofuranosyl-(1→4)-[α-L-rhamnopyranosyl(1→2)]-β-D-glucopyranosyl-β-D-chacotriosyl-26-O-β-D-glucopyranoside (8) [23], aethioside A (9) [24], spongipregnolosides A (10) [25], hypoglaucin H (11) [26], parispseudoside B (12) [24] by using the same set of methods as the new compounds.” in the revised manuscript.

Point 3: In my opinion, when studying cytotoxic properties, data on a normal cell line should be added for control.

Response 3: Thanks for your advice to help us improving the manuscript. The cytotoxic activity of all compounds against normal human NHA astrocytes have been added in Table 3. In fact, we added NHA cells as controls in each experiment, and each time the results showed that the drug had no significant cytotoxic activity (> 50 μM), but due to our negligence, this data was not included in the manuscript initially. Our research group has been engaged in the study of Paris saponins against glioma for a long time, so only normal human NHA astrocytes were used as the control in each experiment. In this experiment, we also used other tumor cells (human cervical cancer cells HeLa, human hepatic cancer cells HepG2 and human pancreatic cancer cells Capan-2) for cytotoxic activity screening. Unfortunately, we only wanted to expand the screening scope and did not conduct in-depth research on other tumor directions, so we did not use normal cells for comparison. Therefore, we only added normal human NHA astrocytes data in the manuscript, hoping to get your understanding.

Point 4: Unfortunately figure 5 is completely unreadable. Might be worth splitting it into separate parts. And also, higher quality of images is needed.

Response 4: Thanks for the constructive comments and suggestions. The original “Figure 5” has been split into separate parts to make it clearly to be seen and all figures have been replaced by high quality images in the revised manuscript.

Point 5: I also want to advise the authors to carefully review the entire text of the article and correct minor typos.

Response 5: Thank you for kind comment. The grammatical errors and typos in the whole text have been corrected as far as possible.

Reviewer 2 Report

It will be good to label the carbon positions - especially to help the reader know which numbers (Carbon and Hydrogen) is being referred to on the HSQC, NOESY, COSY experiments and also on Table 1 and Table 2. At a minimum, it would be good to label the two structures on Page 6 (Figure 4) Line 173.

Author Response

Point 1: It will be good to label the carbon positions - especially to help the reader know which numbers (Carbon and Hydrogen) is being referred to on the HSQC, NOESY, COSY experiments and also on Table 1 and Table 2. At a minimum, it would be good to label the two structures on Page 6 (Figure 4) Line 173.

Response 1: Thanks for your advice to help us improving the manuscript. The carbon positions have been numbered in Figures 1-4 in the revised manuscript to make it more readable and understandable.

Round 2

Reviewer 1 Report

Dear authors!

Thanks for the work you've done. Good luck!